# Sarcasm Detection over Social Media Platforms Using Hybrid Auto-Encoder-Based Model

Dilip Kumar Sharma [1], Bhuvanesh Singh [2], Saurabh Agarwal [3,*], Hyunsung Kim [4,*] and Raj Sharma [5]

1 Department of Computer Engineering and Application, GLA University, Mathura 281406, India
2 Graduate Software Programs, University of St. Thomas, St. Paul, MN 55105, USA
3 Amity School of Engineering & Technology, Amity University Uttar Pradesh, Noida 201313, India
4 School of Computer Science, Kyungil University, Gyeongsan 38428, Korea
5 SRM Institute of Science and Technology, Chennai 603203, India
* Correspondence: saurabhnsit2510@gmail.com (S.A.); kim@kiu.ac.kr (H.K.)

**Abstract:** Sarcasm is a language phrase that conveys the polar opposite of what is being said, generally something highly unpleasant to offend or mock somebody. Sarcasm is widely used on social media platforms every day. Because sarcasm may change the meaning of a statement, the opinion analysis procedure is prone to errors. Concerns about the integrity of analytics have grown as the usage of automated social media analysis tools has expanded. According to preliminary research, sarcastic statements alone have significantly reduced the accuracy of automatic sentiment analysis. Sarcastic phrases also impact automatic fake news detection leading to false positives. Various individual natural language processing techniques have been proposed earlier, but each has textual context and proximity limitations. They cannot handle diverse content types. In this research paper, we propose a novel hybrid sentence embedding-based technique using an autoencoder. The framework proposes using sentence embedding from long short term memory-autoencoder, bidirectional encoder representation transformer, and universal sentence encoder. The text over images is also considered to handle multimedia content such as images and videos. The final framework is designed after the ablation study of various hybrid fusions of models. The proposed model is verified on three diverse real-world social media datasets—Self-Annotated Reddit Corpus (SARC), headlines dataset, and Twitter dataset. The accuracy of 83.92%, 90.8%, and 92.80% is achieved. The accuracy metric values are better than previous state-of-art frameworks.

**Keywords:** autoencoder; BERT; LSTM; sarcasm detection; social media platforms; USE

## 1. Introduction

We now live in the social media era. It has revolutionized the world in terms of communication. Social media opens a world of new possibilities; for example, people may now express themselves with only a finger tap. It is widely used for sharing comments, opinions, support, and sentiments over any topic or image shared over the social media application. Sarcasm is widely used day to day over social media platforms such as Twitter and Facebook. Sarcasm is a turn of phrase for conveying comic, disdain, or bad feelings via exaggerated linguistic constructions. It is a sort of fake politeness used to increase anger inadvertently. Sarcasm can be looked like thinly concealed unkindness. The sarcastic comments and tags are mainly toward political parties and celebrities as they are supposed to be the influencers. Sarcasm has a link to psychological nature such as anxiety and depression. Rothermich et al. [1] showcased that people with depression or medium anxiety during a pandemic used more sarcasm in their interactions over social media. According to a new study, teases feel their statements are less painful than their victims [2]. However, in reality, they are more hurtful.

Sarcasm may easily be detected in a face-to-face conversation by observing the speaker's facial expressions, tone, and gestures. However, identifying sarcasm is challenging because none of these indicators is readily present in written communication. Identifying sarcastic comments for images, videos, or text shared over social platforms is even more difficult as context lies with the image or the main text/comment/headline [3,4]. Sarcasm identification in online communications from social media sites, discussion forums, and e-commerce websites has become essential for fake news detection, sentiment analysis, opinion mining, and detecting of online trolls and cyberbullies [5–8]. Detecting sarcasm is a hot topic of research in current times.

### 1.1. Challenges in Sarcasm Detection

A massive volume of data has considerable potential for corporations to learn more about people's opinions, sentiments, and other aspects. However, there is also a slew of difficulties. For example, sarcasm mainly has positive words, but the context is different, making them negative sentiments [9]. These subtle difficulties have led to wrong assessment of the reviews of product/service in the review analysis or wrong classification in fake news detection. Such difficulties have piqued many organizations and scholars interested in pulling out accurate information from the textual data with sarcasm. Many techniques in NLP are being proposed, considering the contextual part while training the sarcasm detection. The contextual part is learnt from the proximity of words. Different techniques have different proximities and are more related to different context training.

A few examples of sarcastic tweets are provided in Figures 1–3. Figure 1a,b are examples of many sarcastic tweets about politicians and celebrities that are commented on without other linguistic features such as hashtags, emoticons, or quotation marks. They have positive words such as "huge admirer", "love" etc., but the context is different. Figure 2a,b show examples of sarcasm and positive words, but linguistic features such as hashtags, exclamation marks, and question marks are present. Figure 3a,b show sarcastic comments with images. Here the comments are in context to the image. Thus, multiple types of sarcastic comments on social media can be observed, and each one needs to be handled for greater accuracy in its detection. Another issue with social media platforms is that people often use slang or abbreviations in their comments due to their size limit. Understanding these abbreviations is difficult. Moreover, understanding the context of this slang and abbreviations is another challenge.

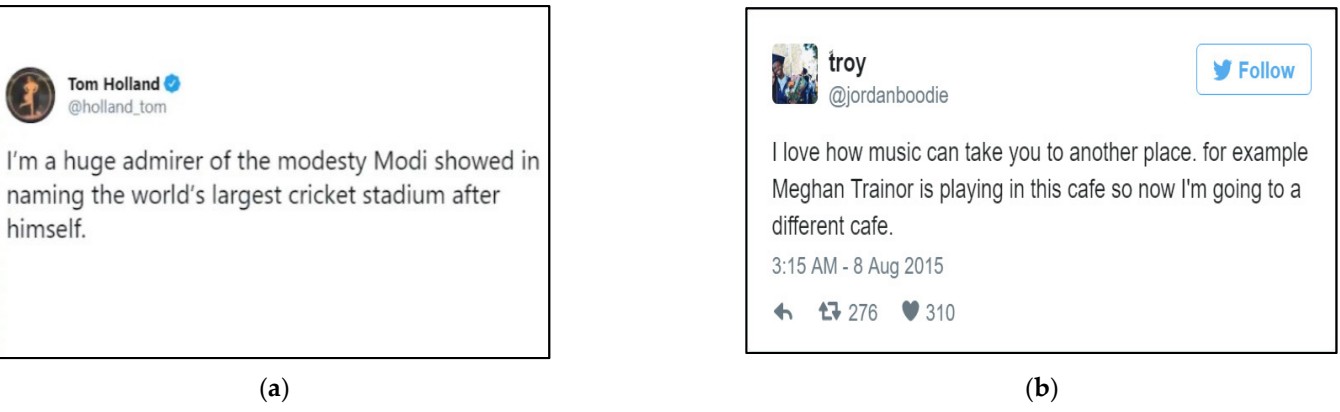

(**a**)　　　　　　　　　　　　　　　　　　　　　(**b**)

**Figure 1.** (**a**) Tweet about politicians [10]. (**b**) Tweets about celebrities [11].

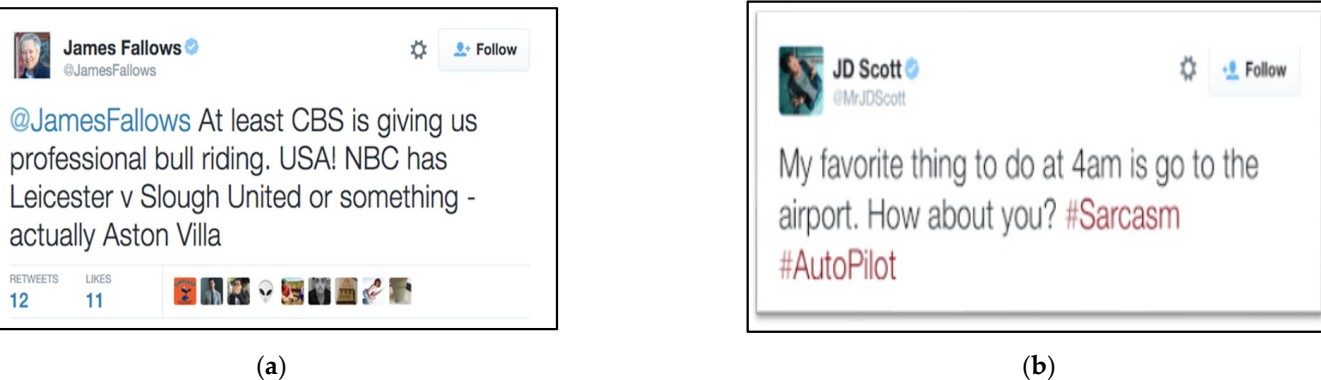

**Figure 2.** (**a**) Sports tweet no hashtags [12]. (**b**) Positive with hashtags [13].

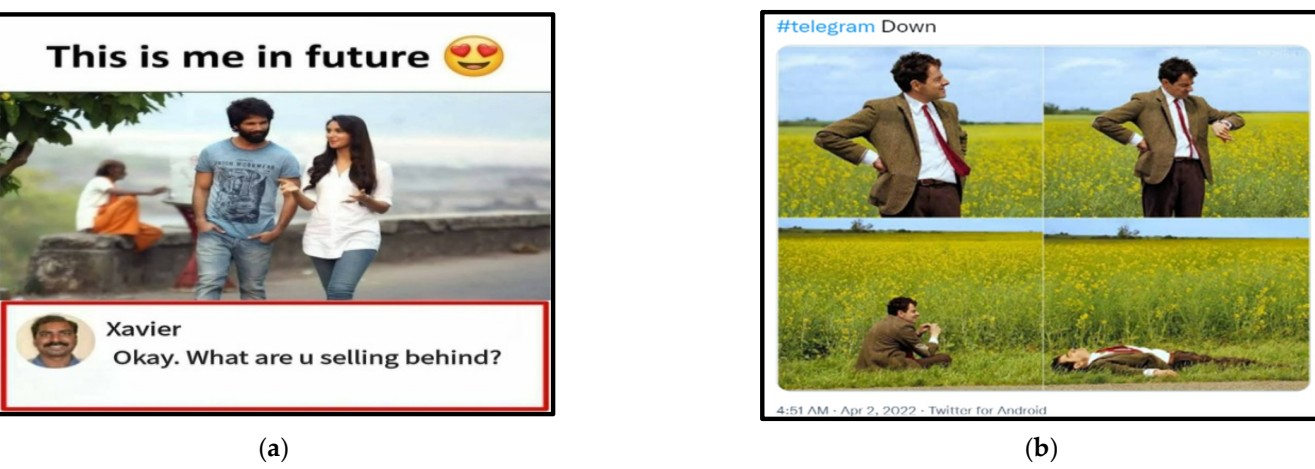

**Figure 3.** (**a**) Images with text [14]. (**b**) Images text and hashtags [14].

So, the main three challenges in sarcasm detection can be attributed to understanding the context, as sarcasm has both positive and negative words. Second, dealing with multiple content types, hashtags, emoticons, exclamation marks, question marks, slang, abbreviations etc. Third, overcoming the limitation of word-based techniques as the context of the sentences play a vital role, and thus word-embedding does not cater to all diverse content types. The third challenge of diversity in the context and content of sarcastic textual data is a significant concern. Each technique has a unique way of understanding and learning the context, so multiple techniques must be applied and assembled into a single framework to handle the diversity. Various models should be selected based on covering short, long, and out-of-context textual data.

### 1.2. Major Contribution

Researchers have attempted to resolve these problems using content- or context-based features. The resolution of the above challenges is required to improve the detection of sarcasm over social media platforms. To resolve these multiple issues, a hybrid approach should be proposed. Each previous research utilizes word-based or sentence-based content or context techniques. Their limitations can be overcome if we employ multiple sentence-based techniques and resolve the issues using a self-learning autoencoder. This thought has inspired the authors of this paper to propose a hybrid approach. Employing multiple branches of BERT with their independent parameters will not help much. The BERT is a good technique where the sentences are a bit longer and have some co-relation concerning the sentences before and after. On tweets data, only BERT will not make a significant improvement as tweets are short and many times correlation between tweets does not exist.

This paper proposes a hybrid model that gathers learning/identification from three sentence-based models: bidirectional encoder representations from transformer (BERT), universal sentence encoder (USE), and unsupervised learning long short term memory-based autoencoder. Each of the three is a sentence encoder/embedding generator. BERT and USE are a transformer-based model that considers the sentence's context, not words, and thus differs from word-based context. Initially, the autoencoder is trained over the datasets. Later, the embeddings from the trained LSTM-based autoencoder and BERT and USE are passed to autoencoder for classification. The learning from these is looped back to autoencoder for further self-correction in an unsupervised way. The accuracy scores of 83.92%, 90.8%, and 92.80% on three publicly available social media-oriented datasets are achieved. These accuracies are better than the previous frameworks proposed earlier by researchers.

The following are the critical goals of this paper:

- Creating a reliable and effective hybrid autoencoder-based model to detect sarcasm on social sites. The model employs a LSTM-based autoencoder for further learning from the results.
- The novelty lies in combining hybrid models from sentence-based embeddings and unsupervised learning utilizing the autoencoder to overcome their limitations. The models are selected by using ablation method and selecting those models which can cover the diverse range of real-time datasets.
- The model is tested against publicly available real-world social media datasets such as Twitter and Reddit. Validation on the real-world datasets is true measure of the metrics for the model. The models can be employed directly over real-world social media contents and can handle diverse contents.
- The model's application is universal and can be used on diverse social media platforms. Datasets have diversified short tweets from Twitter, short sentence from Reddit, and long sentences from newspaper lines.

The practical and commercial applications of the proposed framework are many. Corporations and scholars may use the suggested approach to correctly identify their customers' genuine opinions and emotions from their reviews of their products. This will help them weed out sarcastic comments initially labelled as positive from earlier approaches. Political party cells can be employed in getting the true sentiments of the people and not be inaccurate in understanding due to sarcasm. If there are any learning or strategies to be implemented based on the comments from the public, the political parties can design correct strategies. Fact-checking industries which put manual effort into fake news detection can utilize the model in their internal research and avoid any research over the sarcastic comments. Nowadays, the corporation/company employees are very vocal in sharing their reviews/views about the company over online platforms. The company can re-look into these sarcastic comments and hint about some wrong decisions or plans considered for the employees. Sarcastic comments also lead to inaccurate results of open surveys regarding any political, social or cultural topics. Weeding out sarcastic comments will help improve the overall survey results analysis.

The rest of this paper is laid out as follows. The second section examines relevant research on sarcasm using different techniques. Section 3 outlines the proposed model framework explaining its components. Section 4 shares information about the datasets, experimental results, and comparative analysis with other models. Section 5 concludes and provides direction toward future work.

## 2. Related Work

Although sarcasm has been researched in the social sciences for decades, researching the models to detect sarcasm in texts automatically is a relatively recent topic. Automatic sarcasm detection has recently piqued the interest of researchers in both the machine learning (ML) and natural language processing (NLP) domains [15]. An NLP-based method employs language characteristics and a linguistic corpus to comprehend qualitative infor-

mation. On the other hand, ML methods employ supervised and unsupervised learning techniques to comprehend sarcastic sentences based on tagged or unlabeled material.

Eke et al. [16] reviewed various previous research on sarcasm detection. According to this review article, the most widely utilized feature extraction approaches were n-gram and part-of-speech tagging (POS) techniques. However, for feature representation, binary representation and term frequency were employed. It also observed that information gain and Chi-squared test were widely employed for feature selection. "support vector machines" (SVM), "random forests" (RF), maximum entropy and Naive Bayes classification algorithms were utilized more. Sarsam et al. [17] also reviewed various "adapted machine learning algorithms" (AMLA) and "customized machine learning algorithms" (CMLA) used in research on sarcasm detection. Their findings were similar to Eke et al. [16] Their research showed that "using lexical, pragmatic, frequency, and part-of-speech tagging can contribute to the performance of SVM, whereas both lexical and personal features can enhance the performance of CNN-SVM". Khodak et al. [18] created a large-scale corpus for sarcasm detection. They carried out manual annotation first and later compared their results with techniques such as a bag of words, sentence embeddings, and bag-of-bigrams. They observed that the manual detecting sarcasm was better than other techniques. They said that machine learning techniques can definitely be improved, starting with the use of context to determine sarcasm more accurately.

Machine learning-based models were proposed earlier, and they primarily extract language features and train them over machine learning classifiers. Keerthi Kumar and Harish [19] used machine learning on content-based features. They utilized "mutual information" (MI), "information gain" (IG), and chi-square for feature selection methods and passed it to the clustering algorithms for further filtering. The "support vector machine" (SVM) is used for classification at the final stage. Pawar and Bhingarkar [20] employed machine learning classification models for sarcasm detection on a similar concept. They collated features related to sentiment, punctuations, semantics and syntactic (like number of interjections, unusual words, laughing expression), and patterns. All these feature sets were learned over SVM and random forest for classification.

Another approach was to move above words and learn the context of the sentences. These could be done primarily by using "long short-term memory" (LSTM), bidirectional LSTM or gated or guided attention modules. Ghosh and Veale [21] employed neural network architecture to detect sarcasm over Twitter tweets. They designed a framework comprising CNN and bidirectional LSTM. Two inputs embeddings were provided; one embedding was generated from Twitter data, and the other stream had contextual data about the author (Tweeter) of the tweet. The embeddings were passed through CNN layers for feature learning and then to bidirectional LSTM. The output vectors from bidirectional LSTM were passed to dense layers, and the SoftMax layer carried out the classification. Similarly, Ghosh, Fabbri, and Muresan [22] used various LSTMs with contextual data to identify the sarcastic comments. They analyzed the previous comment to understand the context of the present comment. The context understanding is helpful before predicting whether it is sarcastic or not. Xiong et al. [23] proposed a novel method with a combination of self-matching words and a bidirectional LSTM framework. In self-matching words, the words within the sentences were matched to determine standard information. They used a low-rank bilinear pooling technique to concatenate inconsistencies and composition information to account for possible information redundancy without compromising the classification results. Liu et al. [24] utilized only the content features such as "part of speech" (POS), punctuations, numeric data, and emoticons to detect the sarcasm in the Twitter. Misra and Arora [25] utilized bidirectional LSTM followed by an attention module to detect sarcasm. The bidirectional LSTM provides contextual information considering the previous and subsequent sentences; at attention, the module complements the LSTM by providing the relevant weights for the words. They also created a new dataset, headlines from the website onions (www.theonion.com, accessed on 21 February 2022) and HuffPost's. Akula and Garibay [26] proposed a multi-headed self-attention framework to classify sarcastic

comments over various social media platforms. It also involves gated recurrent units to identify the far-off correlation of words output from the self-attention module. Another multi-headed attention model using bidirectional LSTM was proposed by Kumar et al. [13] Attention mechanism along with gated recurrent unit (GRU) was suggested by Kamal and Abulaish [27].

The transformer-based approach is also another way of learning the context. Babanejad et al. [28] proposed a contextual features-based BERT model for detecting sarcastic comments. Another transformer-based model RCNN-Roberta was proposed by Potamias et al. [29]. They added the utilized RoBerta transformer, a slimmer version of BERT-base and employed bidirectional LSTM. The uniqueness they applied was concatenating the embeddings from the RoBerta and bidirectional LSTM and passing it to the pooling layer. Sundararajan and Palanisamy [30] employed a feature ensemble model with a rule-based approach for sarcasm detection. Another deep learning ensemble model was employed by Goel et al. [31]. Du et al. [32] observed that analyzing the context, including feelings of messages that respond to the target language text and the user's expressive habit, is necessary for identifying sarcasm. They proposed a two-stream CNN, which evaluates both the semantics and emotional context of the target language text. They employed SenticNet to supplement the "long short-term memory" (LSTM) model. The attention mechanism is then used to account for the user's expressive habits. Parameswaran et al. [33] suggested a combination of a machine learning classifier and a deep learning model to retrieve the target of sarcasm from the text. First, they employed machine learning to categorize sarcastic phrases and evaluate if they contain a target (LSTM). The target is extracted using a deep learning model from aspect-based sentiment analysis.

Researchers employed other modalities such as user behavior, hashtags, emotions, and personality traits to detect sarcasm. Garcia et al. [34] used emoticons/emojis to detect sarcasm. Different emoticons were used in sarcastic comments than in other tweets. Yao et al. [35] employed a very novel technique. They utilized four different text-context-based text and Twitter images for sarcasm detection. They used tweets, images in tweets, text over images and image captions. These multi-modalities are learnt over a multi-channel interactions model based on gated and guided attention modules. Hazarika et al. [36] proposed content and context-based embedding methods for sarcasm detection over social platforms. The model employs user features that include stylometric and personality features of users. Illic et al. [37] proposed a framework that employs character-level feature representations of words. It is based on "embeddings from language models" (ELMo). Agrawal, An, and Papagelis [38] proposed a novel emotion-based framework for sarcasm detection. They approach the goal of sarcasm detection as a sequence classification issue, exploiting the natural fluctuations in distinct emotions over the length of a piece of text to investigate the impact of transitions in affective states. In order to train a model to recognize sarcasm, they first divide the text into smaller chunks, represent each chunk with affective data, and then identify the transitions between emotions in the text. Malave and Dhage [39] proposed a framework to track user behavior patterns, personality traits, and context information for sarcasm detection. Sykora, Elayan, and Jackson [40] suggested a model based on hashtags on social media platforms. It studied the hashtags features and identified the sarcastic sentences. Ding, Tian, and Yu [41] used a fusion of multimodal approaches for sarcasm detection. Their model employed residual connections, and had three model variations based on distinct experimental circumstances comprising a multi-level late-fusion learning framework. They used MUStARD dataset which has text, audio, and visual data of sarcasm. They trained each modality in different streams and concatenated in the end following the late-fusion technique. Wen et al. [42] employed a sememe and auxiliary enhanced attention neural model for sarcasm detection. It used auxiliary information to elicit complete comprehension and used Bi-Gru attention model.

Research has also been extended beyond the English language. Techentin et al. [43] studied the sarcasm of native and non-native English speakers. They identified that certain experience features play a role in the ability of non-native speakers to identify and use

sarcastic cues. Farha and Magdy [44]. Al-Hassan, and Al-Dossari [45] researched sarcasm detection in Arabic. Similarly, Swami et al. [46] created and proposed a model for sarcasm detection over Hindi–English tweets.

*Current Issues*

The issue with the above previous research is that most of them target a singular approach. Employing singular content or context-based approaches has its limitations. They might not be a universal solution as context and content change based on the social platforms. There are some research where multiple feature ensemble models have been proposed, but those suggested models combine the modalities from all features correctly. Using features other than content and context does not seem a practical solution. User habits and personality traits are limited to specific individuals and cannot be generalized to create a scalable solution over social platforms.

The authors intend to resolve the issues mentioned above by designing a hybrid model that considers sentence contexts (using transformers). The solution's novelty is utilizing the autoencoder as unsupervised learning for self-correction, which balances the limitations of the individual approaches based on the rules. The solution is universal as multiple techniques are employed, and their learnings are weighted accordingly. The universality lies in the fact that it is validated on diverse datasets such as short comments from the Twitter dataset, long sentences in the headlines datasets and text over images in Reddit/Twitter dataset.

## 3. Proposed Model

The proposed model is a hybrid model where three sentence-based techniques are utilized to provide the surest answer for the sarcasm detection algorithm. The model employs autoencoder, BERT, and USE to identify whether the tweet/text is sarcastic or not. The classification output from the three mentioned techniques is passed to the dense layers, which learn the concatenated sentence embeddings for classification probability. The vectors are concatenated and passed to SoftMax for final classification of input as sarcastic or non-sarcastic.

The flowchart of the proposed model is explained in Figure 4. Initially, the text input is passed to the pre-processing layer. Here the text is pre-processed, and all hangtags and emoji are removed. For the image, the text is extracted from the image using optical character reader (OCR) API pytesseract. Tesseract is google API for OCR. The implementation of tesseract API in python is pytesseract.

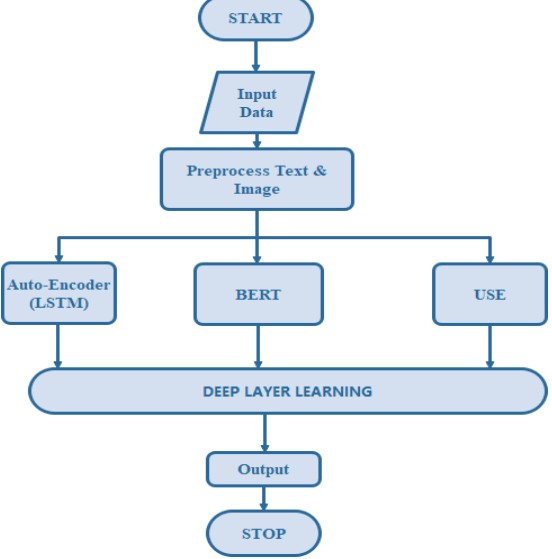

**Figure 4.** Flowchart of the proposed model.

After the text is pre-processed, the pre-training phase begins for the autoencoder. The autoencoder is an unsupervised learning encoder that is based on LSTM. The vector representation generated from the pre-training phase is collected. In the second phase, the initial pre-processed text is passed to BERT and USE models to generate sentence embeddings. All three components output a vector space of embeddings, but as all have different methods of creating embeddings, the embeddings are different. The embeddings generated from the autoencoder, BERT, and USE are concatenated in the third phase. The fused embeddings are passed to dense layers for feature vector learning. The output from the dense layer is passed to SoftMax for the final classification of sarcasm or not sarcasm.

Figure 5 is a detailed illustration of the proposed model. We can see how the text input is passed to the pre-training phase in the LSTM and is also passed to BERT and USE models. The fused sentence embeddings are learnt over dense layers. The probabilities of all three are passed to the final SoftMax layer. BERT, USE, and autoencoder all produce sentence embeddings. It considers the entire sentence and produces a vector. It also considers the previous and prior sentences for producing context-driven sentence embedding. BERT is the latest sentence embeddings generator. BERT works in a bidirectional way, while USE and LSTM-Autoencoder work unidirectionally.

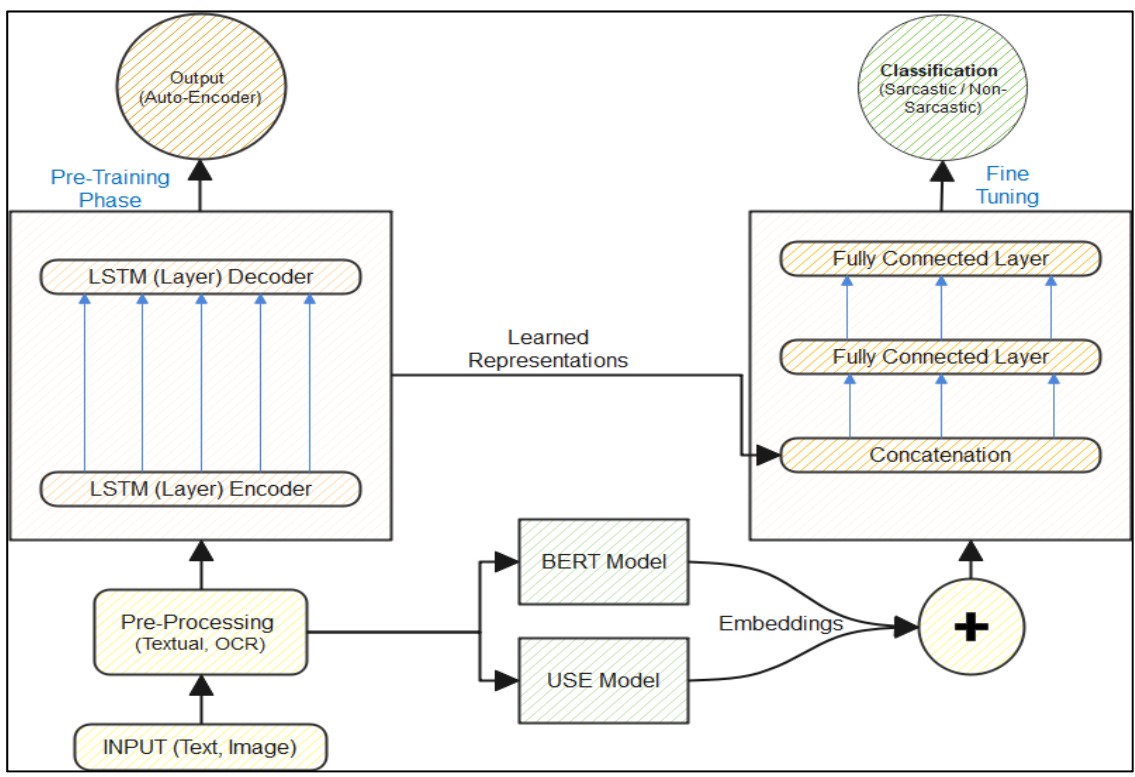

**Figure 5.** Architecture diagram of the proposed model.

### 3.1. AutoEncoder

An autoencoder is a neural network in which the input and output layers have the same values. The fast growth of unsupervised learning methods, where autoencoders find many applications, is responsible for their current prominence.

It comprises an encoder unit buried layer and a decoder unit in its basic form. The encoder aims to convert data input into a lower-dimensional representation known as code. In addition to dimensionality reduction, the decoding side is learned to reduce prediction error. An autoencoder is a standard feed-forward neural network that computes loss function gradients using the backpropagation technique despite its architectural style. Only one class of observations can be used to train an autoencoder, i.e., the rich in training cases non-fraudulent class. This model will learn to recreate typical user actions with low

reconstruction and high reconstruction errors for fraudulent, unknown activities. One technique to employ an autoencoder in a multi-class classification issue is to train several one-class autoencoders and then stack them at the end. After the first training step is completed, a second classifier is created on top of it, utilizing prediction mistakes as input and genuine labels as output.

The encoder and the decoder are the two pieces of the autoencoder. The encoder learns to understand the input and compress it to a bottleneck layer-defined internal representation. The decoder reproduces the input using the encoder's output (the bottleneck layer). We only maintain the encoder once the autoencoder has been taught, and we utilize it to compress input samples to vectors produced by the bottleneck layer. Figure 6 shows the LSTM-based autoencoder architecture. The middle block of "encoded representation" illustrated is the embeddings concatenated with BERT and USE model. Instead of using a bottleneck layer the same size as the input, we would not compress the input's input in this initial autoencoder. This should be a simple problem that the model will learn almost completely, and it will be used to ensure that our model is appropriately implemented.

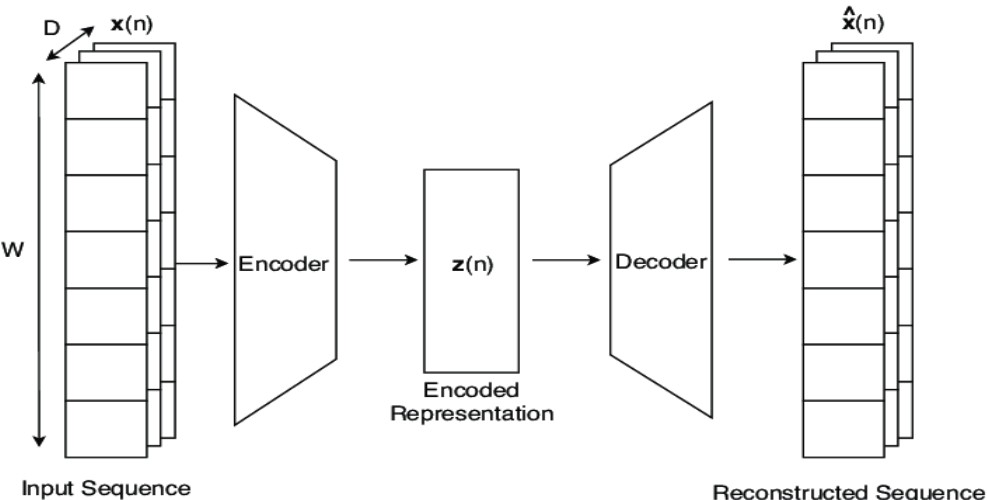

**Figure 6.** The LSTM-based autoencoder [47].

### 3.2. Universal Sentence Encoder (USE)

Text is encoded into high-dimensional vectors using the universal sentence encoder, which may be used for text classification, semantic similarity, clustering, and other natural language applications.

The model is geared for material that is longer than a word, such as sentences, phrases, or short paragraphs. It is trained on various data sources and tasks to dynamically accommodate a wide range of natural language comprehension tasks. The input is a 512-dimensional vector, and the output is a variable-length English text. The outcomes of applying this model to the STS benchmark for semantic similarity may be observed in the sample notebook. A deep averaging network (DAN) encoder is used to train the universal-sentence-encoder model.

On a high level, the goal is to create an encoder that can summarize every text into a 512-dimensional embedding. We utilize the same embedding to tackle various tasks and update the phrase embedding based on the mistakes it makes. Because the same embedding must do several generic tasks, it will only capture the most useful information while ignoring the noise. The hope is that this will lead to a general embedding that can be applied to a wide range of NLP tasks, including relatedness, clustering, paraphrase detection, and text categorization.

There are variants in the universal sentence encoder—transformer encoder and deep averaging network.

In this variation, the transformer encoder employs the encoder component of the original transformer construction. Six stacked transformer layers make up the architecture. A self-attention module is included in each layer, followed by a feed-forward network. The self-attention process considers the word order and the surrounding context when constructing each word representation. To account for the variation in sentence length, the output context-aware word embeddings are added element by element and divided by the square root of the sentence length. As an output sentence embedding, we receive a 512-dimensional vector.

In the deep averaging network, the encoder is based on the design Iyyer et al. [48]. First, the embeddings for all words and bi-grams in a phrase are averaged. The data are then sent into a 4-layer feed-forward deep DNN to produce a 512-dimensional phrase embedding as an output. During training, the embeddings for words and bi-grams are learnt. Figure 7 shows the different variations of the USE model.

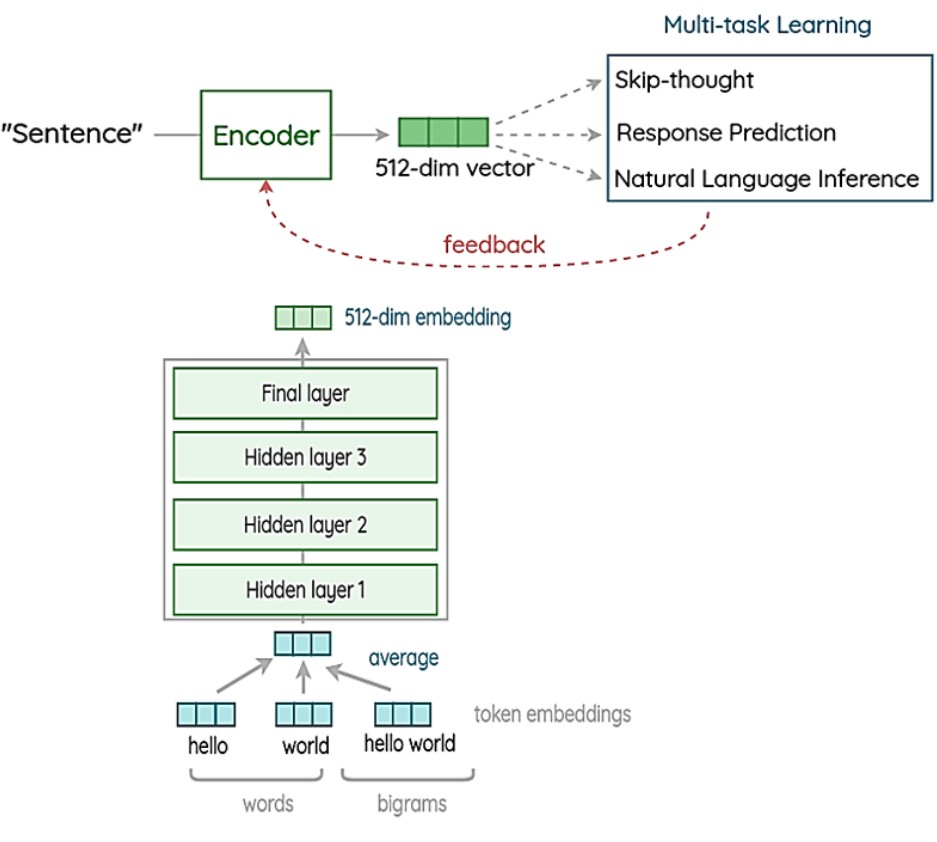

**Figure 7.** Universal sentence encoder multi-task and deep averaging network [48].

In the transfer encoder, because of its complicated design, this encoder provides greater accuracy on downstream jobs but uses more memory and computation resources. Moreover, because self-attention has a temporal complexity that rises with sentence length, the compute time scales drastically. However, it is just slightly slower for short phrases. On the other hand, the deep averaging network has somewhat lower accuracy than the transformer type but has a much faster inference time. The computing time is linear in terms of the input sequence length because we are only doing feed-forward operations.

### 3.3. BERT

BERT is a self-supervised pre-training approach developed and presented by Google that learns to predict purposely hidden (masked) text parts. BERT is based on the architecture of transformers. The research team from BERT-base defines BERT as "BERT stands

for bidirectional encoder representations from transformers [49]. It is designed to pre-train deep bidirectional representations from the unlabeled text by jointly conditioning the left and right context. As a result, the pre-trained BERT model can be fine-tuned with just one additional output layer to create state-of-the-art models for a wide range of NLP tasks".

Previously, natural processing language models were unidirectional, such as GloVe and Word2Vec [50]. They could only move the context window to understand the context in one direction—a sliding window of "n" words (either right or left of a target word) to grasp the context of the target word. BERT, however, can work in both directions. This means the BERT transformers can move the sentences right and left in both directions to fully understand the target word's context. In BERT-base architecture, 12 layers of encoders are stacked together. Because BERT's goal is to develop a language model, just the encoder approach is necessary. It creates a 768-dimensional embedding. In terms of learning, it contains two primary components: "masked language modelling" (MLM) and "next sentence prediction" (NPS). Masked LM (MLM) enables bidirectional training in previously difficult-to-train models. This attribute allows the model to infer a word's context from its surroundings (left and right). In the training phase, NPS is employed. The training method is to feed the model pairs of sentences and learn to predict whether the second sentence in the pair is the following sentence in the original text. During training, 50% of the inputs are a pair in which the second sentence is the sentence after it in the original text, while the other 50% are random sentences from the corpus. The first sentence and the random sentence will be separated. It should be emphasized that BERT-base is a pre-trained model based on a 2500 million word Wikipedia corpus. Figure 8 illustrates the architecture diagram of the BERT-base as described by the creator of the BERT from Google.

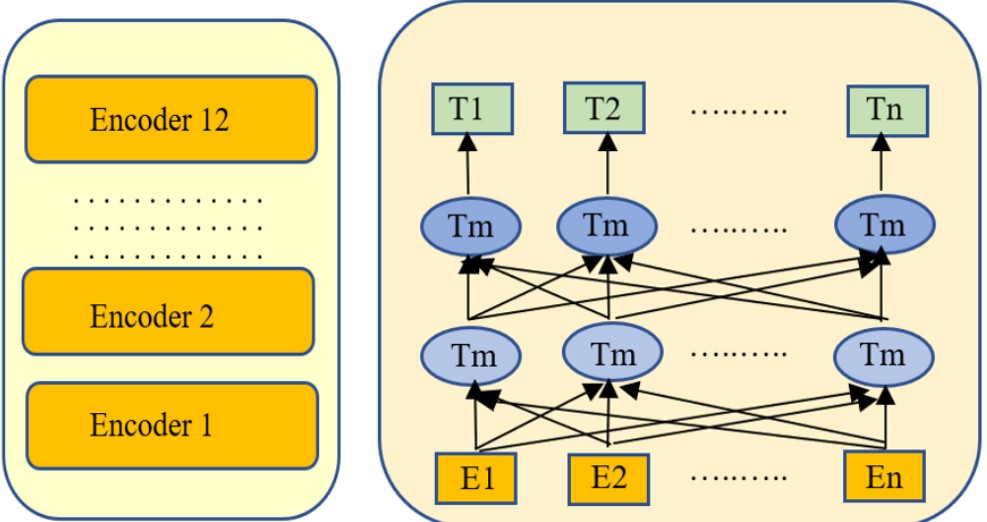

**Figure 8.** The architecture of BERT-base. 12 stacked encoders [51].

## 4. Experimental Results

To assure the model's effectiveness, we conducted an extensive experiment over the three publicly available social networking datasets, Self-Annotated Reddit Corpus (SARC) [18], Twitter dataset [21], and headlines dataset [25].

The model was constructed on the Google TensorFlow platform using the Keras library on a system with 32 GB RAM and GPU Nvidia Quadro RTX 4000 8GB GDDR6. The dense layers had a "Relu" activation function and "Adam" optimizer with a learning rate of $10^{-4}$. The loss function utilized was binary crossentropy, as this is a binary classification problem. These hyperparameters were kept the same across all dense layers for each component. The dataset was randomly divided into three groups, 80% to train and 20% to test. The final

findings were recorded when the maximum accuracy level was achieved. Accuracy metric was employed to stop the network.

### 4.1. Datasets

4.1.1. Twitter Dataset

A Twitter bot called @onlinesarcasm was used to gather tweets for this dataset. This dataset includes tweets and answers to them and the user's mood at the moment of tweeting. The content is the users' tweets/retweets, and the context is the replies to the tweets. 1956 tweets: 895 tweets were acknowledged as sarcastic by their authors, and 1061 were acknowledged as non-sarcastic [21].

4.1.2. SARC Dataset

SARC 2.0, a self-annotated Reddit corpus dataset, comprises comments from Reddit forums. Users' sarcastic remarks, which are with the s token to indicate sarcastic intent, are deleted. We utilize only the original remark in our studies, with no parent or child comments. In our studies, we utilized the "Main Balanced" and "Political" dataset versions, with the latter including sole comments from the political subreddit [18].

4.1.3. Headline Dataset

This news headlines dataset is collected from two news websites: Onion and Huffpost. The onion has sarcastic versions of current events, whereas Huffpost has actual news headlines. Headlines are used as content, and the news article is used as context. It has 26,709 total headlines; 11,725 are sarcastic, and 14,984 are non-sarcastic [25].

### 4.2. Performance Measure

We calculated Accuracy, Precision, Recall, F1score, AUC, and Matthews Correlation Coefficient (MCC) values to evaluate our model's performance. Figure 9 illustrates the confusion matrix. These are the standard performance measure metric for classification problems.

$$Precision\ (P) = TP/(TP + FP)$$
$$Recall\ (R) = TP/(TP + FN)$$
$$Accuracy = TP + TN/(TP + TN + FP + FN)$$
$$F1 = 2 * (Precision * Recall)/(Precision + Recall)$$
$$mAP = \sum_{q=1}^{Q} AvegP(q)/Q$$
$$AUC = \text{Area under the ROC}$$
$$MCC = (TP * TN - FP * FN)/\sqrt{(TP + FP)(TP + FN)(TN + FP)(TN + FN)}$$

**Figure 9.** Confusion matrix.

MCC is statistical correlation value for evaluating models of binary classification. Its responsibility is to evaluate or quantify the difference between the projected and actual values. MCC considers all four values in the confusion matrix. A high number of MCC value indicates that both classes are correctly predicted, even if one class is disproportionately under- (or over-) represented.

Our models' performance on the three publicly available datasets is mentioned in Table 1. The model achieved 83.92%, 90.8%, and 92.80% accuracy scores on SARC, headlines, and Twitter datasets.

**Table 1.** Performance metric of the proposed models on three datasets.

| Dataset | SARC | | | | Headlines | | | | Twitter | | | |
|---|---|---|---|---|---|---|---|---|---|---|---|---|
| **Models** | **Acc.** | **Prec.** | **Recall** | **MCC** | **Acc.** | **Prec.** | **Recall** | **MCC** | **Acc.** | **Prec.** | **Recall** | **MCC** |
| Auto-Encoder | 80.91% | 0.81 | 0.8 | 0.66 | 89.75% | 0.9 | 0.88 | 0.8 | 91.89% | 0.92 | 0.89 | 0.85 |
| USE + Auto-Encoder | 82.46% | 0.82 | 0.82 | 0.67 | 89.90% | 0.91 | 0.86 | 0.81 | 92.32% | 0.93 | 0.92 | 0.85 |
| BERT + Auto-Encoder | 82.42% | 0.83 | 0.81 | 0.67 | 90.35% | 0.91 | 0.88 | 0.81 | 92.40% | 0.94 | 0.91 | 0.85 |
| USE + BERT + Auto-Encoder (Proposed) | 83.92% | 0.83 | 0.85 | 0.68 | 90.81% | 0.92 | 0.912 | 0.81 | 92.80% | 0.95 | 0.91 | 0.86 |

The final model design and selection of sentence embeddings were finalized based on the ablation study. Various combinations of sentence-embedding models were experimented. In all the combinations of mixing models, the uniformity of the parameters was maintained. Similar pre-processing steps were considered. All the various combinations were validated against all datasets. The generic parameters such as activation function, optimizer, loss function, and learning rate were similar in all model combinations. Utilizing the combination of various sentence models experimented in the step-up process, all three datasets have balanced categories of both classes. When the dataset is balanced, the accuracy metric plays a better role in model selection. Table 1 details the performance results of various model combinations on the three datasets. We observed that employing only the autoencoder was not fruitful and led to lower accuracy. Using BERT or USE and autoencoder improved results, but all three throughputs' combinations are the best performance metrics.

The performance measure is better than other previous research on sarcasm detection. Tables 2–4 compare the proposed model results with previous researchers' frameworks on SARC, Twitter, and headlines datasets, respectively.

**Table 2.** Performance comparison of SARC dataset.

| Models | Accuracy | Precision | Recall | F1 |
|---|---|---|---|---|
| CASCADE [36] | 74.00% | - | - | 0.75 |
| SARC [18] | 77.00% | - | - | - |
| Elmo-BiLSTM [37] | 79.00% | - | - | - |
| RCNN-RoBERTa [29] | 79.00% | 0.78 | 0.78 | 0.78 |
| Multi-Head Attn [26] | 81.00% | - | - | - |
| Proposed Model | **83.92%** | 0.83 | 0.85 | 0.84 |

**Table 3.** Performance comparison of the Twitter dataset.

| Models | Accuracy | Precision | Recall | F1 |
|---|---|---|---|---|
| Sarcasm Magnet [21] | - | 0.90 | 0.89 | 0.90 |
| Sentence-level Attn [22] | 74.90% | 0.749 | 0.75 | 749 |
| Self Matching Netwk [23] | 74.40% | 0.763 | 0.725 | 0.744 |
| A2Text-Net [24] | 80.10% | 0.83 | 0.802 | 0.801 |
| Multi-Head Attn [26] | 81.20% | 0.809 | 0.818 | 0.812 |
| Proposed Model | **92.80%** | 0.95 | 0.91 | 0.93 |

**Table 4.** Performance comparison of headlines dataset.

| Models | Accuracy | Precision | Recall | F1 |
|---|---|---|---|---|
| Hybrid [25] | 89.70% | - | - | - |
| A2Text-Net [24] | 86.20% | 0.863 | 0.862 | 0.862 |
| Multi-Head Attn [26] | **91.60%** | 0.919 | 0.918 | 0.918 |
| Proposed Model | 90.81% | 0.92 | 0.911 | 0.915 |

The SARC dataset, which is the largest dataset among the three datasets, has comments from the Redditt websites. From Table 2, we observe that previous scholars primarily used attention mechanisms and LSTM/Bi-directional LSTM in their research [18,27,29,52–54]. Figure 10 shows the confusion matrix of the results obtained on SARC dataset. Figure 11 shows the accuracy comparison chart. Our suggested model uses the BERT-base, which the developers of the BERT-base have demonstrated to be superior to LSTM/Bi-LSTM. Second, Reddit features both reddit and subreddit comments; thus, preceding statements and prior sentences must be interpreted in context. This is not possible with LSTM since it can only forecast the future/forward and not the past/backward. As a result, overfitting is an issue for LSTM. Bi-LSTM, on the other hand, allows for learning from previous and future sequences. However, with Bi-LSTM, the context must be accurately understood; otherwise, the classification will be incorrect. BERT-base, which functions similarly to a transformer, is bidirectional, as is the encoder stack, which performs better in context. BERT-base is also trained set on an extensive corpus with the diverse domain. Thus, our proposed model has better accuracy than previous models that employed LSTM. One of the models utilized by our proposed hybrid model employs BERT-base, which is further fine-tuned with autoencoders.

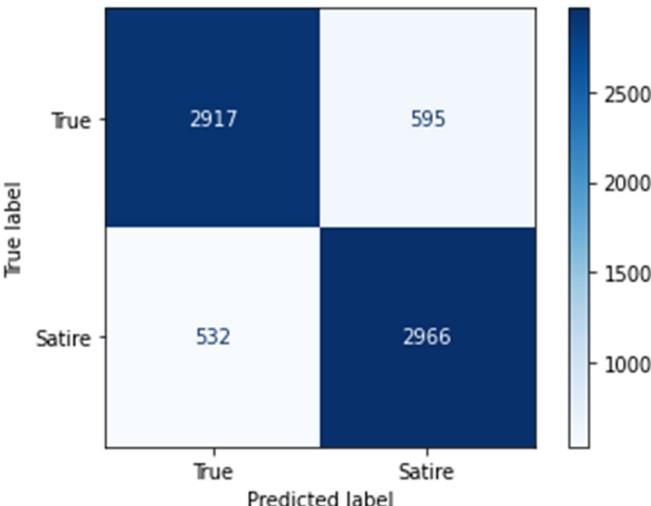

**Figure 10.** Confusion matrix for SARC dataset.

Over the Twitter datasets, an attention mechanism has been employed in most of the previous research [22,23,27,55,56]. Table 3 shows the detailed comparison. Figure 12 shows the confusion matrix results obtained on Twitter dataset. Figure 13 shows the accuracy comparison chart. The attention mechanism is well suited to concentrating on a certain word or sentence. Twitter allows for shorter comments and more use of slang and abbreviation. Because each tweet is unique, the focus cannot be narrowed down. The usage of slang and acronyms may wreak havoc on the attention system. Our proposed model includes an LSTM-based autoencoder that is fine-tuned during the pre-training phase. LSTM better serves short comments/tweets with fine-tuned. The results are relatively comparable on various combinations in all three sentence embeddings models. This is due to the LSTM-based autoencoder, which uses words and performs better on short phrases,

yielding superior accuracy across the Twitter dataset. As a result, the suggested algorithm classifies brief tweets accurately on average, depending on criteria.

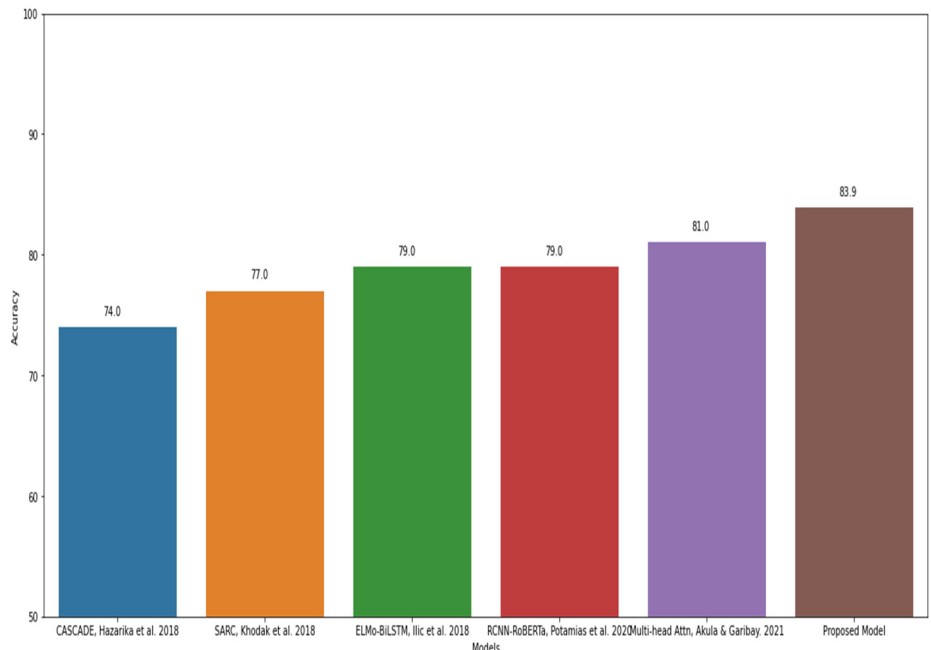

**Figure 11.** Accuracy comparison of previous research [22,23,27,55,56] over the SARC dataset.

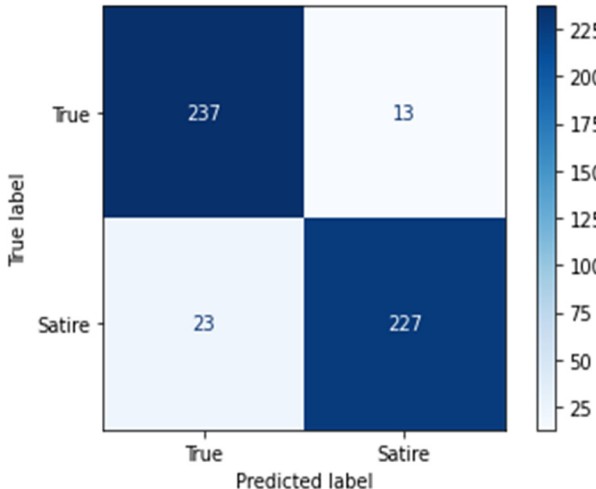

**Figure 12.** Confusion matrix for Twitter dataset.

Over the headlines dataset the proposed model has very good precision and recall values compared to other models. Table 4 provides the detailed comparison results. Figure 14 shows the confusion matrix for Headlines dataset. Figure 15 shows the accuracy comparison chart. Here the accuracy is at par with other previous models because of the range of headlines. There are no prior or subsequent phrases in news headlines, making it difficult to discern the context. Each news headline is unique and can even be caustic. Because no context is associated with the forward and backward sentence embeddings, BERT-base and USE may not operate well with the headlines dataset. The LSTM performs better in this case in recognizing the pattern of specific words used in the headlines. As a result, the precision of our suggested model is comparable to that of other recent study models.

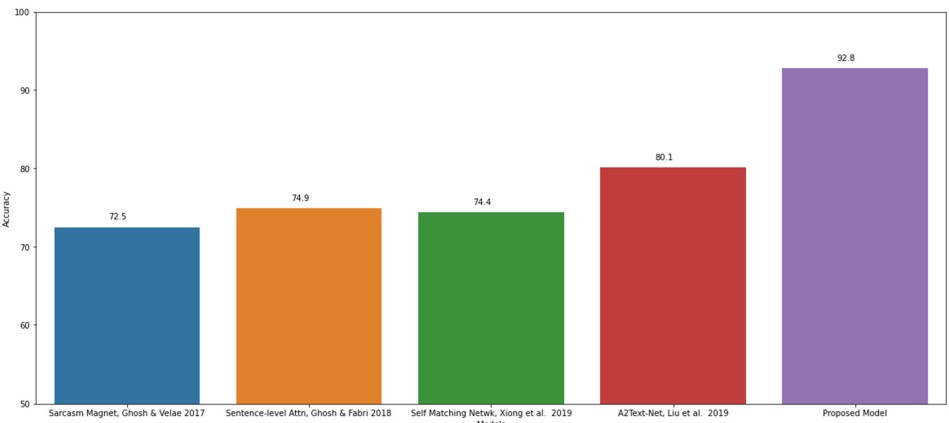

**Figure 13.** Accuracy comparison of previous research [21–24] over the Twitter dataset.

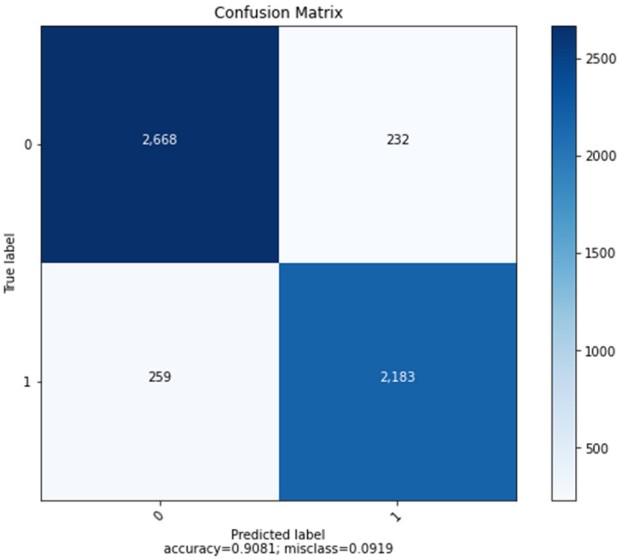

**Figure 14.** Confusion matrix for Headlines dataset.

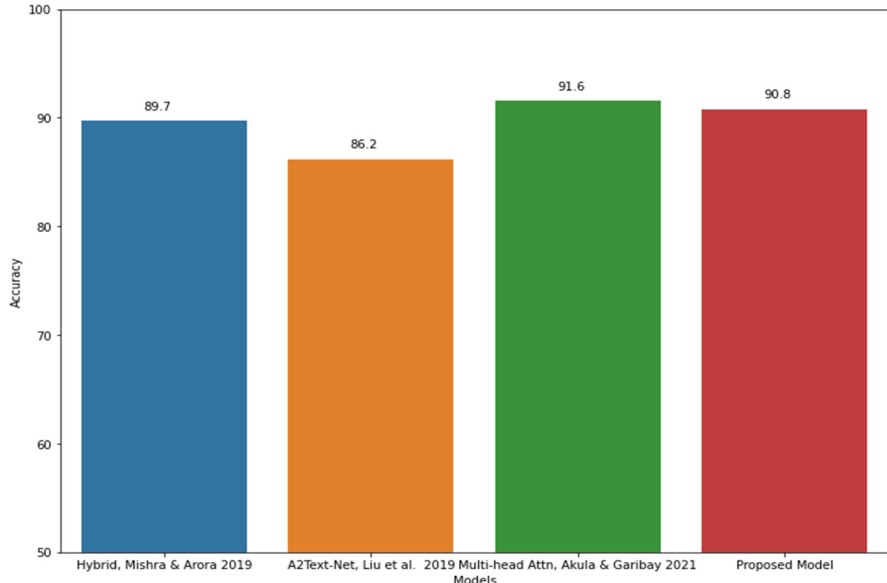

**Figure 15.** Accuracy comparison of previous research [24–26] over headlines dataset.

Figure 16 illustrates the average precision (mAP) information on the three datasets. The area between the precision-recall graphs provides the information on AP. We observe that the AP on SARC datasets is the lowest.

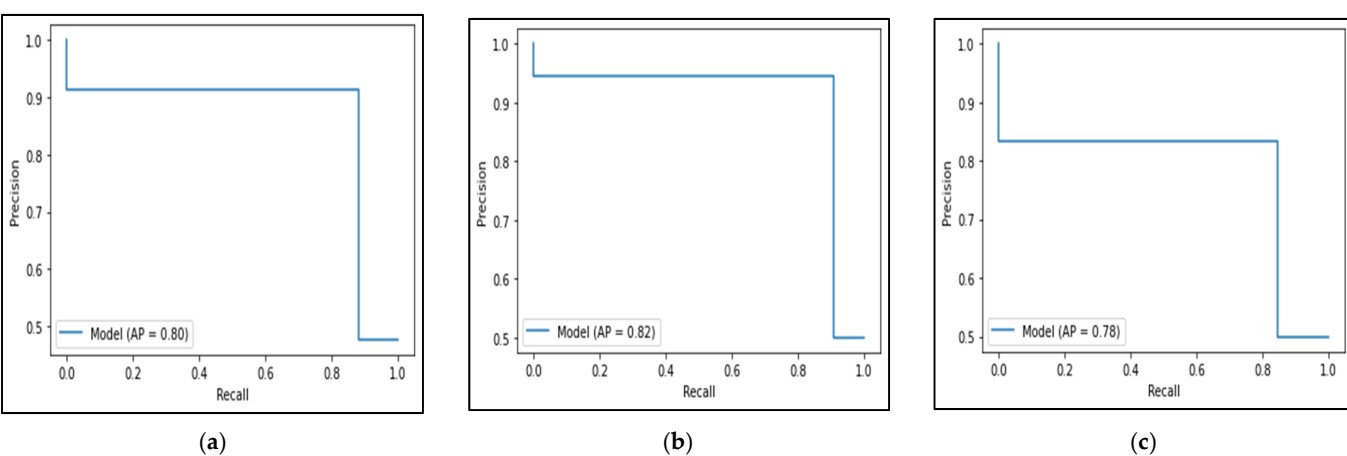

|  |  |  |
|:---:|:---:|:---:|
| (**a**) | (**b**) | (**c**) |

**Figure 16.** Average precision (AP) of (**a**) SARC. (**b**) Twitter. (**c**) Headlines datasets.

### 5. Conclusions

This research paper offers a hybrid model for sarcasm detection combining sentence-based embeddings and autoencoder techniques. The model employs LSTM autoencoder, USE, and BERT-base sentence embedding architecture. The embedding from each technique is learned through the dense layers, and classification probabilities are projected. These projected probabilities are passed to SoftMax for final classification. The framework is universal and performs well on diverse content types. The model is evaluated on real-world social media platforms based on publicly available datasets SARC, headlines, and Twitter. The accuracy achieved by the model is higher than previously state-of-art frameworks in sarcasm detection. The accuracy score of 83.92%, 90.8%, and 92.80% is obtained on the SARC, headlines, and Twitter datasets. The better accuracy is attributed to the utilization of multiple sentence embedding techniques covering various dataset types and fine-tuning autoencoders that balance out the limitations of each technique individually. The model can be used by various corporations, social media mining experts, and fact-checking people to weed out sarcastic comments from the large corpus. The sentiment analysis and opinion mining results will improve after the sarcastic comments are out. There are other linguistic types such as mockery, irony, and pun. These tweets with these language types of sentences can be investigated for further research. Moreover, there must be an improvement over the headline's dataset. A more evolutionary technique such as a genetic algorithm can be looked for further research.

**Author Contributions:** Conceptualization, D.K.S., B.S. and S.A.; methodology, B.S. and S.A.; software, B.S. and R.S.; validation, D.K.S., S.A. and H.K.; formal analysis, B.S. and R.S.; investigation, D.K.S., B.S., S.A. and H.K.; resources, D.K.S. and H.K.; data curation, S.A., H.K. and R.S.; writing—original draft preparation, D.K.S. and B.S.; writing—review and editing, D.K.S., B.S., S.A., H.K. and R.S.; visualization, D.K.S., B.S., S.A., H.K. and R.S.; supervision, D.K.S., S.A. and H.K.; project administration, D.K.S., S.A. and H.K.; funding acquisition, H.K. All authors have read and agreed to the published version of the manuscript.

**Funding:** This work was supported by the Basic Science Research Program through the National Research Foundation of Korea (NRF) funded by the Ministry of Education (NRF-2017R1D1A1B04032598).

**Institutional Review Board Statement:** Not applicable.

**Data Availability Statement:** The datasets used in this paper are publicly available and their links are provided in the reference section.

**Conflicts of Interest:** The authors declare no conflict of interest.

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
