# Peer review of "Sarcasm Detection over Social Media Platforms Using Hybrid Auto-Encoder-Based Model"

_electronics, doi:10.3390/electronics11182844_

Round 1

Reviewer 1 Report

The paper is well written and documented. Organization is good. Presentation is good. Readability is good. Although more detailed considerations on performances would improve the value of the paper, it seems to me suitable to be accepted for publication in this journal.

As noted in the comments below, there are a few issues that would benefit from clarification or further discussion before the paper is Accepted:

1) How to consider the interaction model to guarantee the robust of architecture, the author should give more explanation. Specifically, in the mechanisms applied to the proposed model to avoid overfitting in training, which is common in ML.
2) Also, please rewrite “the most critical parts of this paper” (line 120). I consider this section to be confusing for readers, I suggest listing what would be the strengths and weaknesses of this proposed model versus other works.
3) In adddition, please add the Matthews Correlation Coefficient (MCC) in section 4.2, as this metric is complementary to AUPRC and would strengthen these results. Also, they are useful to evaluate the performance of dichotomous classification models and to validate the robustness of the models with respect to a dataset.

Reviewer 2 Report

The article proposes a sarcasm detection/classification model based on a combination of three deep learning models (LSTM Auto Encoder, BERT, and USE) in a hybrid configuration. This configuration is a kind of ensemble classifier that combines the three models through a fully connected layer, trained to the final classification of sarcasm detection.

In terms of evaluations, three datasets were considered: SARC from Reddit, sarcastic Tweets from [21], and news headlines from [25].   

The authors state that they propose a hybrid method for detecting sarcasm, but in fact, the idea is not innovative, as there are previous works that already do this, e.g., [25]. The approach may be important to reinforce some previous work, but it is not very innovative. Furthermore, although the authors claim that the results are better than previous approaches, this is not entirely true and is the point that concerns me most in this article. For example, when comparing the F1 of their model (0.93) with that of article [21], they take as reference the worst result (0.725) reported there in Table 3 (0.90). They also ignore that in the same table, there are F1 values, from other works, even higher (0.9472). So, there is an omission here that I consider serious. When comparing our proposals with others, we must be as rigorous and transparent as possible!

In terms of writing, the article is well structured a fairly well written, but there are some imprecisions and vague or unjustified/unreferenced claims that appear throughout the paper. Some are just typos, and some are content issues or questions:

106-107: full stop missed. Also, this last sentence would be better rephrased. Sounds a bit strange.

137: “their” => the (?)

151: “Section three” => Section 3 (to be coherent with the subsequent text).

166: “parts of speech” => part-of-speech

178-179: The last sentence sounds strange. Something important from the referred paper is missing here. Of course, "manual detection of sarcasm" is the best approach, since humans still are more intelligent than any AI model/system. This statement should be clarified.

187: “semantics (like interjections)”. Comment: Interjections do not seem to belong to the "realm" of semantics, but more at the lexical level.

195: “one embedding was Twitter data” => one was generated from Twitter data.

195: “authors contextual data” => What does this mean exactly?

208: “in the Twitter tweets” => in Twitter

212: “the website onions” – what is this?

217: “gated RRU” => The term RRU does not appear in the article [27]! Is it GRU?

230: “SenticNet” — Reference is missing! Something is wrong here: SenticNet is a lexical database for sentiment analysis.

248: “Agrawal, An and Papagelis [38] proposed a novel emotion-based framework for sarcasm 248 detection.” — Incomplete related reference. It says almost nothing. At least, you shall identify the main contribution of the work.

258-259: The sentence started with “Residual” sounds strange. It seems that something is missing.

266: “Farha and Magdy [44].” — A reference without any description. Completely inappropriate for a related work.

270: “The issue with the above previous research is that each targets a singular approach.” — this is not accurate!

292: “probability SoftMax logic” — strange/unconventional designation for SoftMax.

346: Gigantic figures does not give good impression. The size should be appropriately balanced: not too small so that it is difficult to read, and not too large ruining the paper aesthetics.

403: “to 403 understand the target word's context fully.” => to fully understand the target word's context.

418: Why did you miss the citation of the original BERT work? You are citing a more generalist source [49] omitting the original work, which does not seem appropriate in a scientific paper.

430: Talking about the Talos library without providing any reference.

481: The F-Score doesn’t appear in Table 1.

556: “and opinions mining” => and opinion mining.

Round 2

Reviewer 2 Report

The authors have addressed all my previous issues and concerns. I have no further comments and agree to proceed to publication.